# An Extended Method for Saccadic Eye Movement Measurements Using a Head-Mounted Display

**DOI:** 10.3390/healthcare8020104

**Published:** 2020-04-21

**Authors:** Youngkeun Lee, Yadav Sunil Kumar, Daehyeon Lee, Jihee Kim, Junggwon Kim, Jisang Yoo, Soonchul Kwon

**Affiliations:** 1Department of Electronic Engineering, Kwangwoon University, Seoul 01897, Korea; yklee1308@gmail.com (Y.L.); sunilyadavsrml@gmail.com (Y.S.K.); jsyoo@kw.ac.kr (J.Y.); 2Graduate School of Smart Convergence, Kwangwoon University, Seoul 01897, Korea; dleogus13813@kw.ac.kr; 3Department of English Language and Literature, Kwangwoon University, Seoul 01897, Korea; kimjihea1206@gmail.com; 4Ingenium College of Liberal Arts, Kwangwoon University, Seoul 01897, Korea; kjg@kw.ac.kr

**Keywords:** developmental eye movement test, eye-tracking, head-mounted display, saccadic eye movement, virtual reality

## Abstract

Saccadic eye movement is an important ability in our daily life and is especially important in driving and sports. Traditionally, the Developmental Eye Movement (DEM) test and the King–Devick (K-D) test have been used to measure saccadic eye movement, but these only involve measurements with “adjusted time”. Therefore, a different approach is required to obtain the eye movement speed and reaction rate in detail, as some are rapid eye movements, while others are slow actions, and vice versa. This study proposed an extended method that can acquire the “rest time” and “transfer time”, as well as the “adjusted time”, by implementing a virtual reality-based DEM test, using a FOVE virtual reality (VR) head-mounted display (HMD), equipped with an eye-tracking module. This approach was tested in 30 subjects with normal vision and no ophthalmologic disease by using a 2-diopter (50-cm) distance. This allowed for measurements of the “adjusted time” and the “rest time” for focusing on each target number character, the “transfer time” for moving to the next target number character, and recording of the gaze-tracking log. The results of this experiment showed that it was possible to analyze more parameters of the saccadic eye movement with the proposed method than with the traditional methods.

## 1. Introduction

Among human senses, vision can be considered the most important one, as we gain the majority of our information regarding the outer world through our eyes [1,2]. In terms of vision, eye movement is critically important, as it decides what information we will obtain through the retina [3]. Eye movement is classified into fixational, gaze-stabilizing, and gaze-shifting movements. Gaze-shifting movement allows the eyes to move voluntarily and simultaneously, and has two different types, smooth pursuit eye movement and saccadic eye movement. Smooth pursuit eye movement is the ability of the eyes to softly follow a moving object, while saccadic eye movement is the ability of the eyes to rapidly move from one object to another [4].

Saccadic eye movement plays a vital role in sports and contributes to athletic performance, as sports involve a high likelihood of continuous movement of the body and objects. Traditional methods for measuring saccadic eye movement include the Developmental Eye Movement (DEM) test or the King–Devick (K-D) test, but these provide only “adjusted time” results. However, if eye-tracking is applied, more detailed information, such as the speed of eye movement, the time the eye stays on the target, and the delay between seeing the target and subsequent action, can be obtained. In addition, eye movement involves natural behaviors, such as head movements, in a real-life environment. Therefore, a measurement system in a virtual reality environment is required. This study proposes an extended method that can calculate the detailed measurements of saccadic eye movement using an eye-tracking module in a virtual reality (VR) head-mounted display (HMD).

## 2. Background Theory

The DEM test is widely used to determine abnormalities related to visual functions, and to identify problems with saccadic eye movement and visual-to-verbal automaticity [5,6]. This type of visual ability is required by athletes in sports that involve continuous movement in response to changes in the surrounding conditions [7]. For the athletes, recognizing and distinguishing any moving object is one of the most crucial factors of their physical ability. Studies demonstrated that saccadic eye movement is an important factor affecting visual functions [8,9,10,11]. Studies related to visual functions have been conducted in many types of sports, such as football, baseball, basketball, and taekwondo. Researches on visual functions have frequently been conducted in baseball players [12]. Researchers also reported that baseball players participating in the Olympic Games and professional baseball players have markedly superior visual functions.

The DEM test can also be used to evaluate reading ability. Eye movements in reading were first described by Javal in 1879 [13]. The DEM test and the modified version (the adult DEM test) have widely been used for testing these visual skills. The authors in [8,14] explained the differences between the DEM test and the adult DEM test. Furthermore, the DEM test can be used to determine the normative values of the DEM test for children speaking different languages [15], and can show the effects of amblyopia in children [16]. It can also be used to determine whether colored overlays could enhance reading performance or scanning ability [17].

Kim et al. [18] introduced a new approach to saccadic eye movement testing, and demonstrated the effectiveness of a virtual reality HMD-based DEM test. The study [18] verified the effectiveness of the saccadic eye movement test, by implementing charts of the DEM test in a VR context, involving a realistic environment with an HMD. This approach simply measured the total time taken to read the chart from start to finish. The method has limitations in providing detailed information factors that can be used to determine eye reading time and action time more accurately.

## 3. Proposed Method

Traditionally, the DEM test is a clinically performed eye movement test used to determine abnormalities in learning-related visual functions, and this test does not involve a realistic environment. The proposed VR HMD DEM test measures saccadic eye movement using the existing DEM test charts in an HMD virtual display. A VR HMD DEM test can calculate the total time taken to read the chart while moving the head and body, enabling detailed measurements of the saccadic eye movement. Eye-tracking technology provides information regarding where a person is looking, what was ignored, and how the pupil reacts to different stimuli. To measure the eye reading time and action time, we proposed a new approach that can calculate the rest time and transfer time for each character in an HMD virtual environment. Our new proposed method also provides an eye movement-trajectory file for each test, which facilitates the evaluation of the user’s input in real-time.

### 3.1. HMD and Eye-Tracking Module

There are many eye-tracking tools available for measuring eye movement activity. Among others, FOVE 0 technology provides a novel approach to eye-tracking and a better estimation of the gaze point. Our proposed method used FOVE 0 eye-tracking technology for measuring the eye movement of a person wearing an HMD, using unity version 2018.3.0f2. The FOVE 0 eye-tracking virtual reality headset used an infrared camera to track the eye movements. The sensors inside the FOVE 0 HMD tracked the user’s pupil, allowing the user to focus on and interact with any targeted object. This enabled eye-tracking by using infrared sensors with a frame rate of 60-fps (90-fps projected), inside the headset. FOVE 0 virtual reality has ultralow-latency, stereo eye-tracking, a 2560-by-1440 display, immersive position-tracking, a software development kit (SDK) with a low-latency compositor, popular game engine plugins, and existing VR game titles. It uses a unity component trail renderer and collider to obtain the rest time and transfer time. As gaze-tracking is an important parameter of the eye-tracking that estimates gaze direction, FOVE 0 uses a robust 6-point calibration system for better estimation of the gaze point.

A working principle of the HMD eye-tracking tool is shown in Figure 1. The HMD eye-tracking tool contains a headset and a display showing frames in front of both of eyes, to provide three-dimensional (3D) views to the user. Figure 1a shows the operation of the eye-tracking module in the HMD. The eye-tracking module includes left and right eyepieces, along with an infrared light source, located between the display and the user’s eye. It also contains eye-tracking cameras pointed towards the eye-facing surfaces of the respective eyepieces. The hot mirrors positioned between the eyepiece and the display at a certain angle allow the infrared light reflected from the eye to reach the eye-tracking cameras, which are placed at the sides of the user’s face. The eye-tracking cameras detect the infrared light reflected from an eye through the hot mirror, and they collect images of the eye. The HMD also contains left and right displays to show left and right images together in front of the user’s eyes, and provides 3D virtual views.

Figure 1b shows a DEM test chart transplanted to the VR HMD. The numbers were taken directly from the hard copy of the DEM test, and the characters are random numbers that cannot be guessed by a subject. The chart is located at a distance of 2-diopter (50-cm) from the eye, with a size 12-times larger than the original. The enlargement of the chart enables the construction of the environment, which facilitates the eye movements in a wider area, and therefore accurate measurement of parameters is achieved. The chart implemented in the VR HMD follows the head and body movements, so that the characters can always be presented at the fixed position with respect to the eyes, regardless of head or body rotations. Consequently, the head and body movements are allowed, while not being considered as variables of the experiment.

### 3.2. Vertical and Horizontal DEM Test

The Bernell DEM test is an indirect eye movement test that measures the “adjusted time”, which is defined as the total time taken to read the characters in the charts while considering omission and addition. Figure 2a illustrates a hard copy of the Bernell DEM test charts. These charts are widely used for the testing of learning-related visual functions in vision therapy, particularly in the context of the evaluation of patients with disorders of the visual system. This basic eye movement test contains a pretest and three tests for evaluation (Test A, Test B, and Test C). The pretest involves a single horizontal line of numbers, in which the subject’s success in naming the numbers (1–9) is assessed. If he/she succeeds in the pretest, Test A, Test B, and Test C are performed successively.

The proposed method applies modified sets of tests, “vertical test” and “horizontal test”. The vertical test consists of Test A and B, which have 40 number characters for each, and the horizontal test is Test C, which has 80 number characters. Test A and B are combined and performed together to match the total number of characters in the vertical test and the horizontal test. Figure 2b shows the DEM test charts incorporated into the VR HMD. Using this method, the subjects can move their heads and bodies freely while reading the DEM charts in the VR HMD. Figure 2c shows the proposed method for measuring saccadic eye movement in a VR HMD environment. This method can measure not only the adjusted time, but also the “rest time”, which is defined as the duration of the gaze focusing on each target number character, and the “transfer time”, which is the duration of the gaze moving to the next target number character, while reading the incorporated charts in the VR HMD environment.

### 3.3. Rest Time and Transfer Time Measurement

Eye calibration is an essential and important procedure for eye-tracking and is performed for each subject before the experiment is started. During calibration, the HMD measures different features of the user’s eye and utilizes these to calculate an accurate gaze point. As a method of calibration, eye gaze telemetry for heat mapping is applied [19]. A gaze vector is calculated first by measuring the position of the pupil center and eye rotation upon images of an eye. By crossing the gaze vector and screen coordinate, the position of the gaze point is estimated, and adjusted by calculating the difference with the position of actual gaze point through a robust 6-point calibration.

A pretest is performed after the calibration and the subjects are asked to read the characters on the line aloud and as quickly as possible, while maintaining a constant distance between the eyes and the display screen. On successful completion of the pretest, the subjects are directed to read the number characters in the vertical test and the horizontal test, consecutively. For each test, the participant’s voice is recorded and the adjusted time is calculated by using Equation (Equation 1) for each test.
(1)$$Adjusted\;time=\frac{Measured\;time\times 80}{80-o+a}$$
where the measured time = the total time taken to read the characters horizontally or vertically; 80 = the total number of characters in the vertical or horizontal test; *o* = the total number of characters omitted while reading; and *a* = the total number of characters repeated or added while reading.

The rest time and the transfer time are measured with the interactions of the components in Unity, called “trail renderer” and “collider”. A trail renderer is a component that can be added to any moving object in Unity to make a trail trajectory behind the object as it moves in a scene. In the proposed method, a trail renderer is attached to the “eye cursor”, which is the gaze point estimated by the HMD eye-tracking module. A collider is a virtual material that can be added to an object in Unity for the purpose of collisions. In our approach, the colliders have same dimensions of (0.04 × 0.04) m and the distance between any two colliders is 0.06 m. Whenever a collision occurs between a collider and an object, it calls out a programmed function and an event is generated. Therefore, colliders are attached to each number character in the DEM test charts so that they can generate an event for each collision with the eye cursor.

An event generated by the first collision between a number character and the eye cursor calls out a timer function of the rest time. The timer of the rest time continues measuring the seconds until the end of the collision and records the time from the start to the end of collision to the log. The rest time is initialized to zero and remains until the next collision occurs. If the eye cursor fails to collide with any number character, another event is generated and the timer function is called out. The timer of the transfer time continues measuring seconds until the next collision of the components and records the time of the failing collision to the log. In this way, the rest time and transfer time are calculated separately and there is no time overlap between the operations of timer functions.

The test outputs 78 rest times and transfer times for the vertical test while 79 rest times and transfer times are measured for the horizontal test. The total numbers of the calculated rest times and transfer times are one less than the number of characters in Test A, Test B, and Test C, respectively. Therefore, since the vertical test consists of two tests while the horizontal test contains only one test, the total number of the vertical test is one less than that of the horizontal test. Figure 3 shows an eye movement-trajectory of the DEM test performed with the proposed method. The image is generated to facilitate the evaluation of the subject’s calibration and a gaze point in real-time. Figure 3a shows a vertical test (Test A and Test B) with 40 characters each and Figure 3b shows a horizontal test (Test C) with 80 characters.

### 3.4. Statistical Analysis

The proposed method of the DEM test implemented in the VR HMD environment was compared with the traditional method of the DEM test with a hard copy through statistical analysis. A paired *t*-test was implemented using SPSS software (ver. 18.0 for Windows, SPSS Inc., Chicago, IL, USA). In our method, we applied non-parametric analysis and a *p*-value lower than 0.05 was taken as a significant value for comparing the data of the subjects. The adjusted time, rest time, and transfer time of the traditional DEM test and the proposed DEM test were statistically analyzed. Furthermore, Pearson correlation analysis was implemented to determine whether the relationship between parameters was correlated. Pearson’s r-value was taken as a representing value for the intensity of correlation under the conditions with *p* < 0.05. The intensity of the correlation by the range of the r-value is described in Table 1.

## 4. Participants and Experiments

To statistically evaluate the proposed method of the VR HMD against the traditional method of hard copy, we recruited participants and administered surveys and eye examinations to select subjects with no limitations in performing the DEM test for both approaches. Each participant took part in a short preliminary interview to verify that he/she did not have any history of ophthalmologic disease or mental disorders, and had an eye examination for checking their visual functions under the supervision of an optometrist. The eye examination of six items included visual acuity, pupillary distance, dominant eye, phoria, stereopsis, and suppression. As eye calibration requires an exact matching of the gaze points from the left and right eyes, the subjects with corrected visual acuity >0.8 were selected. We chose participants who had diagnostic values of exophoria between 0 and 7, esophoria between 0 and 2, stereopsis between 40 and 100-s, and no suppression. We excluded individuals who wore glasses, as it is not possible to wear an HMD and glasses concurrently.

Applying the criteria of visual functions and physical conditions, a total of 30 participants were selected and took part in the experiment. The subjects consisted of college or graduate students aged 21 to 26 years, half male and half female. The experiment was conducted on two individuals per day over three weeks and each experiment took 30 min. All participants gave consent to the experiment and were instructed to perform the experimental procedures. We informed the subjects of the purpose of the experiment, while any psychological element, which would affect the measurement of parameters was not presented.

The pretest and the DEM test of the traditional method using a hard copy were performed first in the experiment. After a success on the pretest, a participant read the number characters in the vertical test and the horizontal test of the traditional method with the DEM test booklet distributed by Bernell. The participant’s voice was recorded while reading and the adjusted time was calculated by Equation (Equation 1) for each test. After the tests, we handed out a questionnaire for checking the subjective symptoms of the subject. The subject was instructed to fill in the questionnaire with the quantitative severity of symptoms, which he/she experienced while performing the DEM test of the traditional method.

Table 2 shows a questionnaire of subjective symptoms, which contains 25 items in four categories including inspection-related, physical symptom-related, eye symptom-related, and headache-related symptoms. The items in the questionnaire were taken from the Social Support Questionnaire (SSQ) [20], which is a psychometrical survey used to measure satisfaction. The severity of symptoms was represented with scores from 0: strongly negative to 4: strongly positive.

After a break to reduce the eye strain, the vertical test and the horizontal test of the proposed method were performed after the calibration was completed. We conducted the experiment with the DEM test implemented in the VR HMD environment, and surveyed with the same procedures as the traditional method. Figure 4 shows the experimental environment of the proposed method. During the vertical test and the horizontal test, the subject could move his/her head and body freely. The log files of rest time and transfer time, which were calculated whenever a gaze point made contact with a character, were created for each test. Table 3 shows a sample of the log files, which have 78 rest times and transfer times for the vertical test and 79 rest times and transfer times for the horizontal test. The serial number of 40 was not recorded in the vertical test, as the test had successive experiments of two tests, Test A and Test B.

## 5. Results

### 5.1. Subjective Symptoms

The averages of subjective symptom scores recorded by subjects after the DEM test of the traditional method and the proposed method are compared in Table 4. The *p*-values of 11 items among 25 questions were lower than 0.05 and less than half of the items showed a significant difference between the two tests. Except for Questions 3.7 and 4.2, all symptoms were more prevalent after the DEM test of the proposed method than the traditional method. Figure 5a–d shows the chart comparisons of the subjective symptom scores for Questions 1.1–1.5 (Figure 5a), Questions 2.1–2.7 (Figure 5b), Questions 3.1–3.9 (Figure 5c), and Questions 4.1–4.4 (Figure 5d), respectively.

### 5.2. Experimental Results

The averages of the adjusted time of the vertical and horizontal test of the traditional method and the proposed method are compared in Table 5 and Figure 6. The *p*-value of 0.001 is sufficiently small for the null hypothesis to be rejected and the alternative hypothesis to be adopted that there is a significant difference between the two experimental results. Compared to the traditional method, the adjusted time of the proposed method was approximately 5 s longer in the vertical test and 18 s longer in the horizontal test. Additionally, in the DEM test of the traditional method, the adjusted time of the vertical test was longer than of the horizontal test, while the proposed method outputs the opposite results.

The correlation of the adjusted times between the vertical tests of the traditional method and the proposed method is shown in Table 6. The r-value was 0.530, which indicated a distinctly positive linear relation between the two methods in the vertical test. A scatter plot of correlation in the vertical test is illustrated in Figure 7. In the same way as the vertical test, Table 7 shows the correlation of the adjusted time between the horizontal DEM test of the traditional method and the proposed method. The r-value was 0.575, which also implied a distinct positive linear relation between the two methods in the horizontal test. Figure 8 illustrates the scatter plot of correlation in the horizontal test.

The averages of the rest times and transfer times of the proposed method are compared in Table 8 and Figure 9. The *p*-value of 0.001 was sufficiently small that there was a significant difference between the two experimental results. The rest time of the horizontal test was approximately 7 s shorter compared to the vertical test while the transfer time was 20 s longer. However, the transfer time was longer than the rest time in both of the tests. Figure 9 shows chart comparisons of the rest time (Figure 9a) and transfer time (Figure 9b), respectively, in the vertical test and the horizontal test.

Table 9 shows samples of the averages of the rest time and transfer time from five subjects. In the case of Subject 2 and 5, the rest time was shorter and the transfer time was longer in the vertical test when compared to other participants. Especially, subject 5’s transfer time of the horizontal test was more than four times longer than the one of Subject 4. In addition, only the rest time of horizontal test from subject 5 was longer than the one of the vertical tests.

## 6. Discussion

Several aspects of the DEM test implemented in the VR HMD environment were confirmed through comparing the proposed method with the traditional method. For verification of the system as a DEM test, the subjective symptoms, correlation between the two methods, and record of parameters were measured.

### 6.1. Subjective Symptom Scores

In the survey of subjective symptoms, the difference between the traditional method and the proposed method was statistically significant in terms of dizziness, sensing uncomformity of the characters, and overall discomfort while performing the experiment. The difference was also statistically significant with respect to headaches, shoulder pain, and eye tiredness. There were no other statistically significant differences between the two methods. Except for 2 out of 25 items, the differences of the averages of subjective symptoms were less than 1. The fatigue of an eye was the only item that demonstrated a statistically significant difference and the difference of the score over 1. Therefore, the proposed method of the DEM test using HMD is not expected to enhance the subjective symptoms if a rest for relieving eye strain is adequately taken after the test.

### 6.2. Correlation Analysis

The distinctly positive correlation of the adjusted time between the conventional method and the proposed method in the vertical and horizontal tests, proved the effectiveness of the proposed method as a DEM test. In both of the vertical and horizontal tests, a longer time was required to complete the tests in the proposed method than in the traditional method. However, this is only a numerical difference of measurements, and the measured data of both methods show similar patterns. We assume that the cause of the difference was the enlargement of the DEM charts, since the distance between the two characters was greater in the proposed method than in the conventional method.

### 6.3. Individual Characteristics

The rest time and the transfer time measured with the proposed method show the individual characteristics of saccadic eye movement. Subject 2 and 5 from Table 9, recorded a shorter rest time and longer transfer time than other participants in the vertical test, and this indicated that the reaction rate was rapid while the eye movements were slow, which can imply poor eye-functioning of detection and discrimination of moving objects. In contrast, Subjects 1, 2, and 5 recorded completely opposite measures of rest times and transfer times in the vertical test, and this indicated that the eye movements were rapid while the reaction rate was slow, which can signify a lack of reading ability.

### 6.4. Developments

The effectiveness of the measurements of saccadic eye movement using HMD was evaluated and verified by comparing subjective symptoms, correlation analysis between the traditional method and the proposed method, and individual characteristics. The safety and practicality of the measuring system, and distinctly positive correlation between the two methods were confirmed through statistical and comparison analysis. The VR HMD environment was essentially different from the real-world environment, and we had difficulties in controlling the variables, such as the size of the test chart and the distance between the number characters and the eye. The implementation of measurements of saccadic eye movement using eye-tracking equipment in a real-world environment seemed to have a stronger correlation with the conventional method and make controlling variables easier compared to the method of using HMD in VR environment.

## 7. Conclusions

This study proposed an extended method for saccadic eye movement measurements using a DEM test implemented in a VR HMD environment. Through the proposed method, measurements of different parameters, such as the adjusted time, rest time, and transfer time were possible while an individual was moving his/her head and body. The eye tracking-trajectory was also provided so that an evaluator could supervise the calibration and a progress of the test in real time. The transfer time and the rest time were used to compare whether the eye movements were slower or faster than the action time or vice-versa. Thereby, the test was capable of diagnosing deficits of saccadic eye movement. The approach has the potential for future applications in many fields, such as medicine, industry, driving, etc., where it is necessary to measure saccadic eye movements. It will be especially useful for distinguishing persons at risk of disabilities in reading, character recognition, visual processing, and verbal speed. The proposed method is expected to be highly improved if the technologies of deep learning, voice recognition, and other comfort devices are combined.

## Figures and Tables

**Figure 1 healthcare-08-00104-f001:**
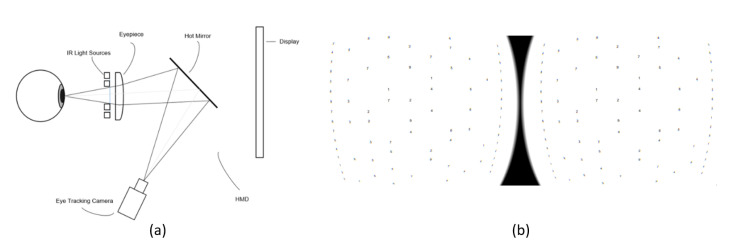
A schematic diagram of the proposed method for measuring saccadic eye movement using a virtual reality (VR) head-mounted display (HMD): (**a**) Eye-tracking module in the HMD; (**b**) The Developmental Eye Movement (DEM) test chart transplanted to the VR HMD.

**Figure 2 healthcare-08-00104-f002:**
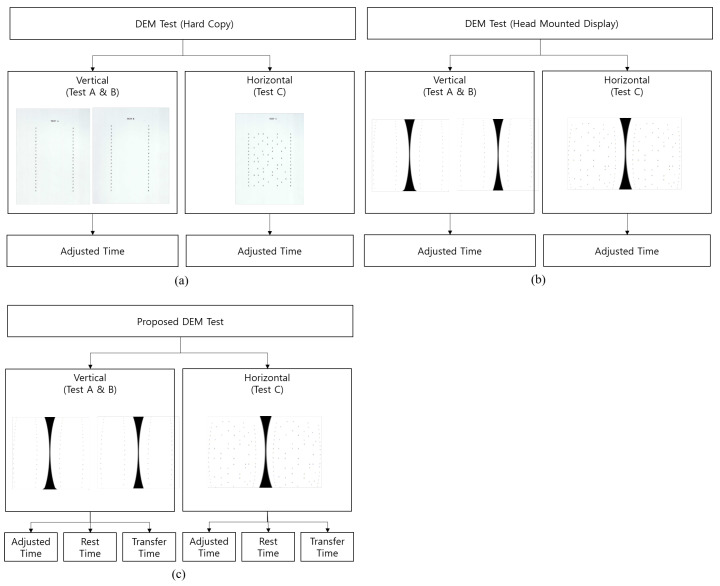
A comparison of the proposed and traditional methods for measuring saccadic eye movement: (**a**) A hard copy of the Developmental Eye Movement (DEM) test charts; (**b**) The DEM test charts incorporated into the VR HMD; (**c**) The proposed method of measurements with the DEM test charts incorporated into the VR HMD.

**Figure 3 healthcare-08-00104-f003:**
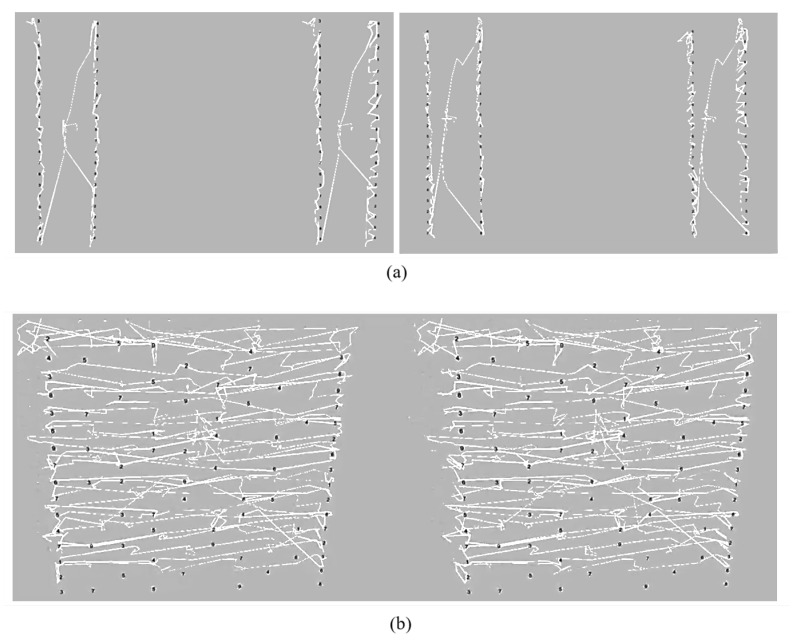
The eye movement-trajectory of the DEM test performed with the proposed method: (**a**) Test A and B; (**b**) Test C.

**Figure 4 healthcare-08-00104-f004:**
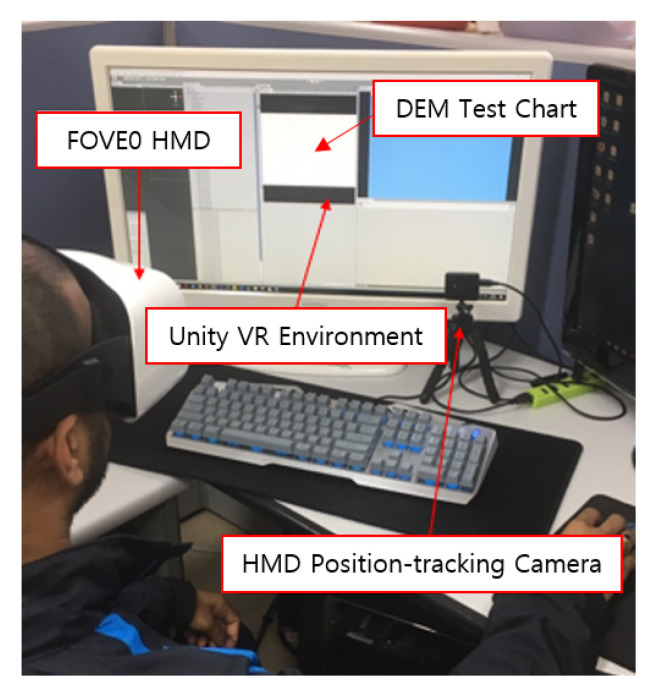
The experimental environment of the proposed method.

**Figure 5 healthcare-08-00104-f005:**
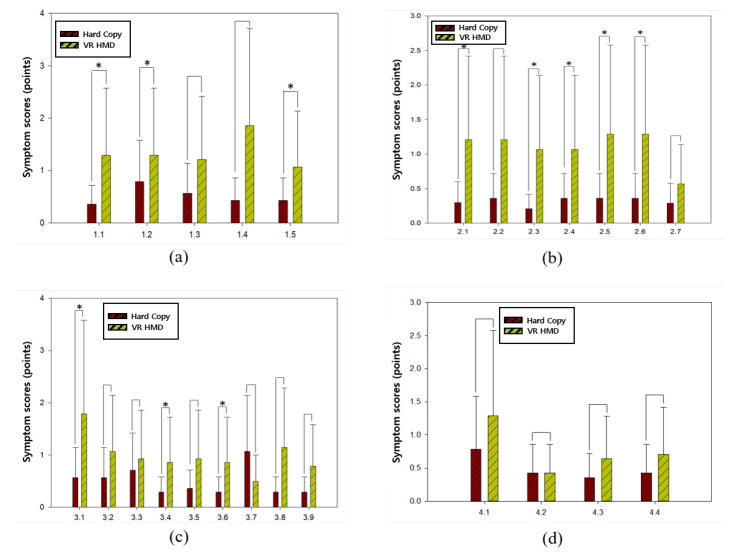
Chart comparisons of the subjective symptom scores (* stands for the items which showed a significant difference between the two tests): (**a**) Symptom scores for Questions 1.1–1.5; (**b**) Symptom scores for Questions 2.1–2.7; (**c**) Symptom scores for Questions 3.1–3.9; (**d**) Symptom scores for Questions 4.1–4.4.

**Figure 6 healthcare-08-00104-f006:**
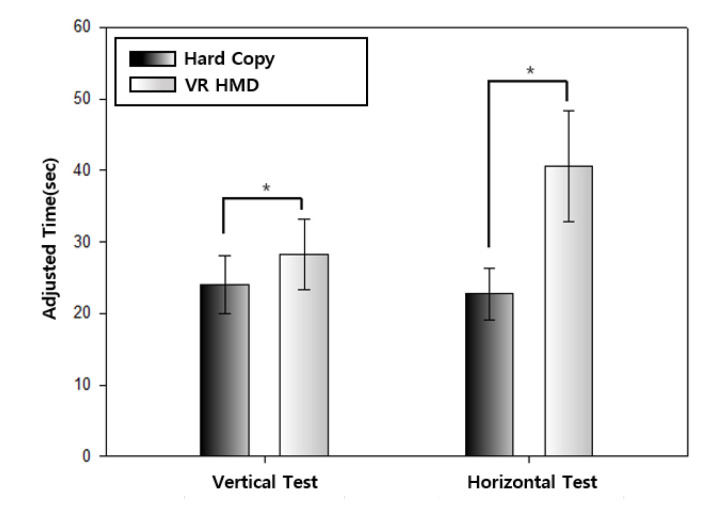
A chart comparison of the adjusted times.

**Figure 7 healthcare-08-00104-f007:**
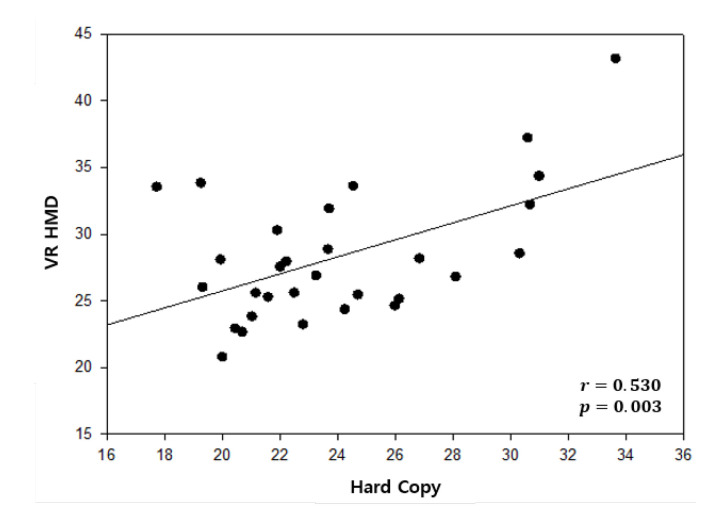
Scatter plot of correlation in the vertical test.

**Figure 8 healthcare-08-00104-f008:**
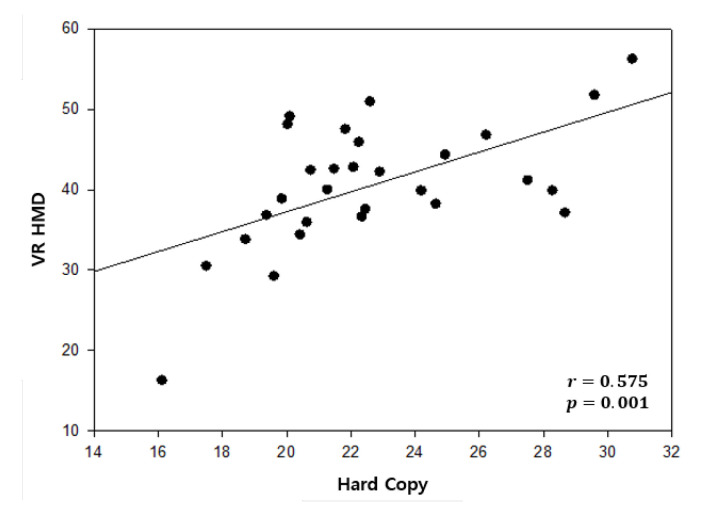
Scatter plot of correlation in the horizontal test.

**Figure 9 healthcare-08-00104-f009:**
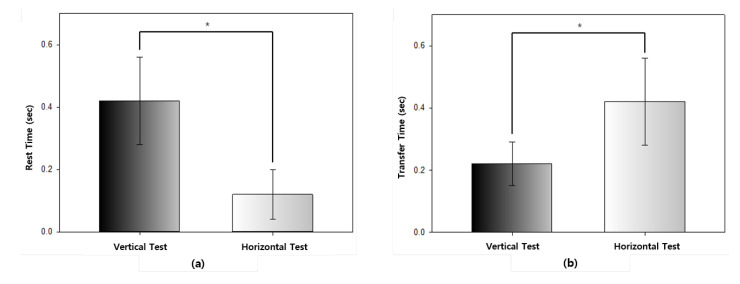
Chart comparisons of the rest time and transfer time: (**a**) Rest time; (**b**) Transfer time.

**Table 1 healthcare-08-00104-t001:** The intensity of correlation by the range of the r-value.

Range of r-Value	Intensity of Linear Relationship
−1.0 to −0.7	Strongly negative (−)
−0.7 to −0.3	Distinctly negative (−)
−0.3 to −0.1	Weakly negative (−)
−0.1 to +0.1	Negligible
+0.1 to +0.3	Weakly positive (+)
+0.3 to +0.7	Distinctly positive (+)
+0.7 to +1.0	Strongly positive (+)

**Table 2 healthcare-08-00104-t002:** Questionnaire of subjective symptoms.

Serial No.	Items
1	1.1. I felt dizzy. 1.2. Image of the target does not combine into one and it appears as several images. 1.3. The screen lacks appropriate sharpness. 1.4. The size of the target was too small to read. 1.5. During the experiment, I was hindered by overall discomfort.
2	2.1. I was physically and psychologically uncomfortable. 2.2. My body was tired. 2.3. My shoulders ached. 2.4. My neck felt stiff. 2.5. I developed a headache during the experiment. 2.6. I felt dizzy. 2.7. I felt nauseous.
3	3.1. My eyes were tired. 3.2. I felt my eye pulling. 3.3. If you felt pulled, where did you feel it? 3.4. My eyes stung. 3.5. My eyes watered. 3.6. My eyes were broken. 3.7. My eyes trembled. 3.8. After watching things, objects are looked up. 3.9. Surroundings seemed cloudy.
4	4.1. I felt headache in the front of my head. 4.2. I felt headache at the back of my head. 4.3. I felt headache on the top of my head. 4.4. I felt headache on the side of my head.

**Table 3 healthcare-08-00104-t003:** A sample of the logs of the rest times and transfer times of the vertical test and the horizontal test.

Serial No.	Vertical Test			Horizontal Test		
	Character No.	Rest Time (s)	Transfer Time (s)	Character No.	Rest Time (s)	Transfer Time (s)
1	3	0.10	0.53	2	0.03	0.07
2	7	0.03	0.20	5	0.03	0.11
3	5	0.10	0.10	9	0.03	1.09
4	9	0.19	0.03	4	0.07	0.04
5	8	0.19	0.20	3	0.06	0.29
⋮	⋮	⋮	⋮	⋮	⋮	⋮
38	3	0.28	0.09	4	0.04	0.05
39	6	0.12	0.07	6	0.06	0.38
40	4	-	-	3	0.10	0.51
41	6	0.09	0.06	6	0.14	1.15
42	3	0.12	0.03	3	0.05	0.21
⋮	⋮	⋮	⋮	⋮	⋮	⋮
76	3	0.37	0.49	3	0.03	0.47
77	7	0.25	0.18	7	0.07	0.43
78	5	0.46	0.19	5	0.03	0.45
79	9	0.44	0.03	9	0.17	1.86
80	8	-	-	8	-	-

**Table 4 healthcare-08-00104-t004:** Subjective symptom scores.

Questions	Hard Copy	VR HMD	t	*p*-Value
	Mean ± SD (Points)	Mean ± SD (Points)		
1.1	0.36 ± 0.49	1.29 ± 0.99	−2.56	0.010
1.2	0.79 ± 0.97	1.29 ± 8.25	−2.11	0.035
1.3	0.57 ± 0.87	1.21 ± 0.97	−1.7	0.086
1.4	0.43 ± 0.64	1.86 ± 0.77	−1.89	0.058
1.5	0.43 ± 0.64	1.07 ± 0.99	−2.16	0.030
2.1	0.3 ± 0.63	1.21 ± 1.12	−2.43	0.015
2.2	0.36 ± 0.63	0.79 ± 1.12	−1.56	0.119
2.3	0.21 ± 0.42	1.07 ± 0.91	−2.43	0.015
2.4	0.36 ± 0.63	1.07 ± 0.91	−2.43	0.026
2.5	0.36 ± 0.63	1.29 ± 0.99	−2.4	0.016
2.6	0.36 ± 0.49	1.29 ± 0.99	−2.5	0.010
2.7	0.29 ± 0.46	0.57 ± 0.85	−1.30	1.940
3.1	0.57 ± 0.75	1.79 ± 1.05	−2.63	0.009
3.2	0.57 ± 0.64	1.07 ± 0.99	−1.73	0.083
3.3	0.71 ± 0.91	0.93 ± 1.26	−0.53	0.590
3.4	0.29 ± 0.46	0.86 ± 0.77	−2.27	0.023
3.5	0.36 ± 0.63	0.93 ± 0.99	−1.99	0.460
3.6	0.29 ± 0.46	0.86 ± 1.09	−1.99	0.049
3.7	1.07 ± 1.14	0.50 ± 0.76	−1.58	0.114
3.8	0.29 ± 0.46	1.14 ± 0.94	−2.36	0.480
3.9	0.29 ± 0.46	0.79 ± 0.80	−1.38	0.100
4.1	0.79 ± 0.89	1.29 ± 1.20	−1.38	0.168
4.2	0.43 ± 0.51	0.43 ± 0.51	0.00	1.000
4.3	0.36 ± 0.49	0.64 ± 0.74	−1.19	0.234
4.4	0.43 ± 0.64	0.71 ± 0.72	−1.41	0.157

**Table 5 healthcare-08-00104-t005:** A comparison of the adjusted times.

	Traditional Method Using Hard Copy	Proposed Method Using VR HMD	t	*p*-Value
	Mean ± SD (s)	Mean ± SD (s)		
Vertical Test	23.99 ± 4.08	28.29 ± 4.92	−5.32	<0.001
Horizontal Test	22.70 ± 3.60	40.61 ± 7.75	−15.33	<0.001

**Table 6 healthcare-08-00104-t006:** Correlation between the vertical tests of the traditional method and the proposed method.

	Traditional Method Using Hard Copy	Proposed Method Using VR HMD	*p*-Value
Traditional Method using Hard Copy	1		0.003
Proposed Method using VR HMD	0.530	1	

**Table 7 healthcare-08-00104-t007:** Correlation between the horizontal tests of the traditional method and the proposed method.

	Traditional Method Using Hard Copy	Proposed Method Using VR HMD	*p*-Value
Traditional Method using Hard Copy	1		0.001
Proposed Method using VR HMD	0.575	1	

**Table 8 healthcare-08-00104-t008:** A comparison of the rest time and transfer time.

	Vertical Test	Horizontal Test	t	*p*-Value
	Mean ± SD (s)	Mean ± SD (s)		
Rest Time	0.19 ± 0.04	0.12 ± 0.08	6.44	<0.001
Transfer Time	0.22 ± 0.07	0.42 ± 0.14	−11.11	<0.001

**Table 9 healthcare-08-00104-t009:** Samples of the rest times and transfer times from five subjects.

Sample No.	Vertical Test		Horizontal Test	
	Avg. Rest Time (s)	Avg. Transfer Time (s)	Avg. Rest Time (s)	Avg. Transfer Time (s)
1	0.353	0.191	0.120	0.322
2	0.177	0.218	0.104	0.405
3	0.293	0.220	0.129	0.436
4	0.182	0.133	0.095	0.318
5	0.168	0.209	0.399	0.413

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
