# Peer review of "An Extended Method for Saccadic Eye Movement Measurements Using a Head-Mounted Display"

_healthcare, 2020, doi:10.3390/healthcare8020104_

Round 1

Reviewer 1 Report

76: High frame rate: 60fps cannot longer be considered as a high frame rate, please delete 'high'

87: What is a hot mirror?

Figure 1: There is further explanation needed for (b), e.g. what do the numbers mean?

Figure 2: Same as figure 1.

112: The eye calibration needs further explanation. What exactly was measured and how was the tracking calibrated using the measurements?

117: Which test?

Equation 1: Where does the 80 comes from?

Figure 3: The line - background differentiation is really hard for readers with not so good vision. It would be better to have a black and white image.

151: If you only recruited participants without limitation in performing DEM test, from where did you get the a-priory knowledge of their ability? Or did you recruit everyone and then did the interview and selected your participants?

How did you decide wether or not they have no limitation?

156: What do you consider as normal visual acuity? 

156: Did you exclude participants with a refractive error or just participants who wear glasses? (so you included eyes with uncorrected refractive error?)

General question: Did you have ethical approval? What was the rational to NOT let the participants know what the aim of the study was?

171: What was the score? Is this a validated method? here should be evidence provided for the validity of this questionnaire.

173 onwards: highly redundant

Table 2: should be supplementary material (instead a table with a descriptive analysis of the sample not for each patient).

188: The description of the statistical analysis should be in methods not in the results.

194: Pearson does not test if parameters are independent!

Why did you chose Pearson? If you have two methods testing the same, why did you expect to find a low correlation by any chance? Here ICC or any other agreement measure would be most suitable, but Pearson doesn't tell you anything other than both measure eye movement. 

231: the discussion is insufficient. There is barely any discussion about the study limitations and the results.

251 onwards: How can you conclude that from your results?

In general, this is an interesting article with a lot of potential interest for some readers, but needs major revision.

Reviewer 2 Report

There are major problems with the manuscript as presented. Key ones are that dynamic visual acuity is repeatedly discussed, but the DEM has nothing to do with this. Visual acuity (VA) relates to the minimum character size which can be resolved--DVA examines this during head (or occasionally target) motion. But all the stimuli here were of fixed size. In addition, when a paper stresses the importance of body motion, you'd expect it to be controlled, but it wasn't. Subjects may have moved, or may not have, but we don't know how. And, finally, what was the point of doing this in VR? A conventional head-mounted eye tracker could have been used while the subjects sat in a swivel chair and rotated while holding the DEM cards. And while there is excessive detail in the results, the discussion section is the most cursory I've ever seen in a manuscript.

I'll give line number references below for specific comments.

Line 2: 1st problem--the King Devick and DEM tests aren't intended to measure VA! To quote from kingdevicktest.com: "The King-Devick Test is a two-minute rapid number naming assessment in which an in individual quickly reads aloud single digit numbers and evaluates impairments of eye movements, attention and language function." This isn't a good start to the paper. 

Line 18: The references 1 and 2 are actually about the DEM and K-D test in relation to reading ability, not vision per se.

Lines 24-26: In neither the DEM or K-D test is either the target or the subject in motion; they have nothing to do with dynamic visual acuity.

Lines 34 and following: The authors might want to look at the study Ayton, L. N., et al. (2009). "Developmental Eye Movement Test: What is it Really Measuring?" Optometry and Vision Science 86(6). 722-730, which found that DEM performance was uncorrelated with any ocular motor parameters, though it did relate to perceptual aspects of reading. 

Paragraph beginning with line 51: None of the measures cited are a measure of VA. There seems to be a redefinition of VA in this groups' work.

Paragraph beginning on line 59: How is a K-D test seen in a VR headset more realistic than one printed on a card or on a monitor? You could shake your head while reading one and, if it was on a card, it could rotate with you easily.

Line 76: 60 Hz isn't a high frame rate. And given the increasing ubiquity of VR, is any of this section needed?

Lines 106-108: Again, how does this measure DVA if the optotype size is fixed?

Line 109: None of the 3 types of time are defined before being used repeatedly.

Line 127: If it's a VR construct, the collider isn't a physical material.

Lines 133-136: Is rest time what's usually called fixation duration or dwell time in the eye movement literature?

Line 141: Is transfer time the saccade duration?

Paragraph beginning on line 149: Was informed consent ever obtained? It's not mentioned.

Lines 177-178: It states "During the vertical test and the horizontal test, the subject could move his/her head and body freely." So motion was undefined? Or not even present? What does this tell anyone? In DVA studies motion is a controlled variable. This seems at odds with the stated point of the study.

Table 2: Here and in following results, are the individual values needed in a table? Why not a graph with data points, mean and error bars?

Table 3: Why is this table here?

Table 5: Why not just give r and p in the figure and get rid of the table? Same point for table 6.

Table 8: What's special about these 5 subjects?

Line 227: Are the symptom scores higher in the VR tests just because of the well-known "simulator sickness" issues sometimes arising in VR?

Line 231: This is the shortest and least informative discussion I've ever seen in a full-length manuscript. Maybe the differences between the two presentation methods was because people felt queasy in the VR headset; given that motion was uncontrolled and unmonitored, we have no idea if it had any effect on anything.

Reviewer 3 Report

The paper by Lee et al. provides a nice description of a new technology for dynamic eye movement testing.  It is well written and relatively easy to follow.  The use of eye tracking in combination with the ability to control for head and body movements represent major improvements over previous methods. I agree with the authors that this technology has many important future applications.  Below are a few minor points to consider prior to publishing.

Line 55.  The sentence beginning with “It doesn’t provide…” is somewhat awkward and could be better organized and / or shortened.

The supporting evidence for the new method of DEM testing is substantial, and I am excited about the use of goggle technology for this purpose.  Although correlations were relatively strong between traditional vs. goggle, the absolute discrepancies between the two methods are also significant.  Can the authors go into a little more detail regarding their head movement explanation?  
